# Translation and Validation of the Swedish Version of the Tilburg Frailty Indicator

**DOI:** 10.3390/healthcare11162309

**Published:** 2023-08-16

**Authors:** Amelie Lindh Mazya, Anne-Marie Boström, Aleksandra Bujacz, Anne W. Ekdahl, Leo Kowalski, Magnus Sandberg, Robbert J. J. Gobbens

**Affiliations:** 1Division of Clinical Geriatrics, Department NVS, Karolinska Institutet, 141 83 Huddinge, Sweden; 2Department of Geriatric Medicine of Danderyd Hospital, 182 88 Danderyd, Sweden; 3Theme Inflammation and Aging, Nursing Unit Aging, Karolinska University Hospital, 141 86 Huddinge, Sweden; 4Division of Nursing, Department NVS, Karolinska Institutet, 141 83 Huddinge, Sweden; 5R&D Unit, Stockholms Sjukhem, 112 19 Stockholm, Sweden; 6Department of Learning, Informatics, Management and Ethics, Karolinska Institutet, 171 77 Stockholm, Sweden; 7Department of Clinical Sciences Helsingborg, Lund University, 251 87 Helsingborg, Sweden; 8Department of Health Sciences, Lund University, 221 00 Lund, Sweden; 9Faculty of Health, Sports and Social Work, Inholland University of Applied Sciences, 1081 HV Amsterdam, The Netherlands; 10Zonnehuisgroep Amstelland, 1180 HV Amstelveen, The Netherlands; 11Department Family Medicine and Population Health, Faculty of Medicine and Health Sciences, University of Antwerp, 2610 Wilrijk, Belgium; 12Tranzo, Tilburg University, 5037 AB Tilburg, The Netherlands

**Keywords:** frailty assessment, Tilburg Frailty Indicator, psychometric properties, translation

## Abstract

The Tilburg Frailty Indicator (TFI) is a questionnaire with 15 questions designed for screening for frailty in community-dwelling older people. TFI has a multidimensional approach to frailty, including physical, psychological, and social dimensions. The aim of this study was to translate TFI into Swedish and study its psychometric properties in community-dwelling older people with multimorbidity. A cross-sectional study of individuals 75 years and older, with ≥3 diagnoses of the ICD-10 and ≥3 visits to the Emergency Department in the past 18 months. International guidelines for back-translation were followed. Psychometric properties of the TFI were examined by determining the reliability (inter-item correlations, internal consistency, test–retest) and validity (concurrent, construct, structural). A total of 315 participants (57.8% women) were included, and the mean age was 83.3 years. The reliability coefficient KR-20 was 0.69 for the total sum. A total of 39 individuals were re-tested, and the weighted kappa was 0.7. TFI correlated moderately with other frailty measures. The individual items correlated with alternative measures mostly as expected. In the confirmatory factor analysis (CFA), a three-factor model fitted the data better than a one-factor model. We found evidence for adequate reliability and validity of the Swedish TFI and potential for improvements.

## 1. Introduction

Frailty is an age-related syndrome of reduced capability to cope with minor stressors [1]. The pathophysiological processes that lead to the development of frailty are not yet fully understood, and there is no international consensus on how to define frailty. There are numerous instruments to identify older individuals with frailty and several reviews on the topic [2]. Evidence-based interventions can slow down the progress of frailty [3]. Frailty assessment is highly important due to the coming demographic changes with an aging population.

There are two dominant models of frailty in the literature. The frailty phenotype by Fried et al. describes a biological phenotype and is measured by the presence of at least three of the following criteria: weight loss, slowness, weakness, exhaustion, and low physical activity [4]. The accumulation of deficit model by Mitnitski and Rockwood is operationalized into a frailty index (FI). The ratio between present deficits and possible deficits from a predefined list of symptoms, signs, and diagnoses estimates the degree of frailty [5]. Both models identify older individuals at risk for disability, dependency, hospitalization, institutionalization, and mortality [6]. The Clinical Frailty Scale is developed from the same model as the FI and is a measure based on clinical judgment [7]. It assesses comorbidity, physical function, cognition, and functional level using pictures and written descriptions to stratify patients into one of nine categories ranging from 1 (very fit) to 9 (terminally ill) [7]. 

An alternative approach to frailty emerged when researchers and health professionals grew objections to the one-sided focus of the physical/medical aspects of the previous models of frailty. Concerns were raised regarding a possible compromise of the holistic attention of older individuals that could lead to fragmentation of care [8]. Inspired by the World Health Organization’s (WHO) definition of health as “a state of complete physical, mental, and social well-being and not merely the absence of disease or infirmity”, Gobbens et al. presented a new model of frailty in 2010 [8]. After literature research and expert meetings, their work resulted in the following definition of frailty: “Frailty is a dynamic state affecting an individual who experiences losses in one or more domains of human functioning (physical, psychological, social), which is caused by the influence of a range of variables and which increases the risk of adverse outcomes” [8]. The authors emphasize the multidimensionality of frailty and the need of a multidisciplinary approach to the complex needs of frail individuals. Based on this definition, Gobbens et al. developed the Tilburg Frailty Indicator (TFI), a self-administered questionnaire that was found to be user-friendly for frailty screening in community-dwelling older people [9]. 

The original Dutch version of the TFI has shown good psychometric properties in community-dwelling older people [9]. The TFI has been translated and validated in several countries but not yet in Sweden. Recently, findings from over 25 studies of the psychometric properties of the TFI have been summarized in a review [10]. The authors conclude that the reliability of the TFI as measured by internal consistency and test–retest reliability, as well as the criterion and construct validities, are noteworthy. Two systematic reviews of multi-component frailty instruments concluded that the TFI had the most robust evidence of reliability and validity and was the most studied frailty instrument in terms of psychometric properties [11,12]. The TFI is recommended for screening for frailty in public health and in primary care [13,14]. Most studies have only focused on the construct validity and criterion validity of the TFI. There are only two studies that have presented results from factor analysis [15,16]. A recent systematic review of frailty assessment in community-dwelling older adults by Rasiah et al. highlights the lack of factor analysis for many frailty instruments and their underlying factors [2]. They recommend confirmatory factor analysis of an existing instrument as a path to further explore the construct of frailty and underlying factors. 

Multimorbidity is often defined as having two concurrent chronic diseases, but there are several other definitions [17]. In Sweden, one definition of multimorbidity extends the definition to include three or more chronic conditions and adds frequency of admittance and/or visit to the emergency department, thus creating a definition of a population not only afflicted by multimorbidity but also in need of complex health care support [18]. Early identification of frailty in people with multimorbidity is recommended since it enables individually tailored interventions and thus reduces the risk of adverse outcomes [19]. There are no studies on the performance of the TFI in community dwelling older people with multimorbidity [10]. 

The aim of this study was two-folded. The first aim was to translate and adapt the Tilburg Frailty Indicator to Swedish. The second aim was to assess the reliability and the validity of the Swedish version of the Tilburg Frailty Indicator in a sample of community-dwelling, older people with multimorbidity, including investigating the correlation of the Tilburg Frailty Indicator to the Clinical Frailty Scale and the Frailty Phenotype (concurrent validity).

## 2. Materials and Methods

The translation from English followed the ten-step procedure recommended by the International Society for Pharmacoeconomics and Outcomes Research (ISPOR) [20]: 1. preparation, 2. forward translation, 3. reconciliation, 4. back translation, 5. back translation review, 6. harmonization, 7. cognitive debriefing, 8. review of cognitive debriefing results and finalization, 9. proofreading, and 10. final report [20]. Authorization of translation was obtained from Robbert J J Gobbens in April 2018.

Data from the 24-month follow-up of the Comprehensive Geriatric Assessment with a Mobile Teams Trial (GerMoT) was used. Inclusion criteria were: 1. age ≥ 75 years; 2. ≥3 visits to the Emergency Department (ED) in the past 18 months; 3. ≥ three diagnoses in at least three chapters of the International Statistical Classification of Diseases and Related Health Problems, 10th revision (ICD-10); and 4. community-dwelling (not in a care home). The study protocol has been published, and the study is registered at ClinicalTrials.Gov (NCT02923843) [21]. The study was conducted in a city in Region Skåne, in the south of Sweden. Data were collected through structured face-to-face interviews and clinical assessments by research nurses between October 2018 and June 2020 in the participants’ homes. 

### 2.1. Measurements

The TFI consists of ten questions on determinants of frailty and demographic information (part A). Only part B consists of questions on frailty (fifteen) and is, therefore, included in the psychometric analysis. Questions 11–25 are called items 1–15 hereafter. Physical frailty is assessed by eight items (1–8): physical health, weight loss, walking difficulties, problems with balance, problems with vision and hearing, diminished hand strength, and physical tiredness. Psychological frailty is assessed by four items (9–12): memory problems, depressive symptoms, anxiety, and coping. Social frailty is assessed by three items (13–15): living alone, social relations, and social support. Four items (9–11, 14) have three answer categories: ‘‘yes’’, ‘‘sometimes’’, and ‘‘no’’, the remaining items have only two categories: “yes” and “no”. The range for the total score is 0–15, and a higher score corresponds to a higher degree of frailty. A scoring instruction is provided. Earlier studies used a cut-off of 5 or more for frailty [9]. 

The Frailty Phenotype was measured with a modified version of the five criteria from the Cardiovascular Health Study by Fried et al. [4]. In our study, we use a different measurement for physical activity than in the original study. *Exhaustion* was assessed by two questions from the Center for Epidemiological Studies Depression Scale (“I felt that everything I did was an effort” and “I could not get going”) [22]. *Weakness* was measured with grip strength using a Jamar hand-held dynamometer with two attempts; the best values were used and adjusted for gender and BMI, using the same cut-offs as presented by Fried et al. in 2001 [4]. *Slowness* was assessed by measuring the walking speed over 4 or 5 m with walking aids if needed. A walking speed under 0.8 m/s was considered a cut-off for frailty [23]. For the assessment of *physical activity*, the Swedish version of the International Physical Activity Questionnaire–Short Form (IPAQ-SF) was used [24]. It summates duration and frequency of activity, leading to the categorization of activity levels as high, moderate, or low. A low activity level was considered frail per this criterion. *Weight loss* was estimated by self-reported unintentional weight loss of more than five percent in the past year. Participants were classified as frail if they met three or more of these criteria and non-frail if less than three. The Clinical Frailty Scale was scored as the final instrument in the assessment. The cut-off for frailty was a score of five or more.

### 2.2. Statistical Analysis Reliability and Validity

Inter-item correlations were analyzed using Pearson’s correlations, and a *p* ≤ 0.05 was considered statistically significant. The correlation coefficient was evaluated using the classification of Callegari–Jacques (weak < 0.30; moderate 0.30–0.60; strong 0.60–0.90; very strong ≥ 0.90) [25]. The internal consistency of the items of the TFI and its domains was assessed using an alternative to Cronbach’s alpha, theKuder–Richardson formula (KR-20), due to the data being dichotomous and Mcdonald’s omega. A minimum level of 0.70 was considered adequate [26]. Test–retest reliability after 14 (±2) days was analyzed using Cohen’s kappa and the percentage of agreement for each item. For interpretation of the kappa value, the reference values of Landis and Koch were used (slight < 0.20; fair 0.20–0.40; moderate 0.40–0.60; substantial 0.60–0.80; almost perfect > 0.80) [27]. 

To investigate concurrent validity, the Pearson correlation between the TFI sum score and the Frailty Phenotype and the Clinical Frailty Scale, respectively was performed [4,7]. In order to find evidence for concurrent validity, a strong correlation of 0.6 or higher was expected. Construct validity was analyzed using Pearson correlations of 14 of the 15 TFI items, with established measures that addressed the same construct. Item 13 was not included in the analysis since there was no alternative measure. The alternative measures for the physical frailty domain were: the International Physical Activity Questionnaire-Short Form (IPAQ-SF), measurement of body mass index (BMI), walking speed and balance test from the Short Physical Performance Battery (SPPB), questions on hearing and sight (able to make telephone calls with or without aid/able to read a text with or without aid) used in the validation study of the Chinese TFI), grip strength (measured with a hand-held dynamometer) and a statement on exhaustion from the Center for Epidemiologic Studies Depression Scale (CES-D)-I felt that everything I did was an effort [22,24,28,29,30,31]. For the psychological domain, the measures were the Montreal Cognitive Assessment (MoCA), the Hospital Anxiety and Depression Scale (HADS) and the Pearlin Mastery Scale (PMS) [32,33,34]. The alternative measures for the social domain wereone question on loneliness from the study SWEOLD-Does it happen that you are troubled by loneliness? and the Oslo 3-item social support scale (OSSS-3) [35,36]. It was expected that the score of the alternative measures would at least have a moderate correlation of 0.40 with the items measuring the same domain of frailty in order to provide evidence for construct validity. Data were analyzed using IBM SPSS Statistics for Windows, Version 26.0 and 27.0 (IBM Corp., Armonk, NY, USA) and JAMOVI 1.6.23.

Structural validity was analyzed with confirmatory factor analysis (CFA) in R using the Lavaan package [37]. Diagonally weighted least squares were used as an estimator due to binary items. We compared the model fit of two alternative models. In a one-factor structure-fitting, all items as one factor were compared with the three-factor structure suggested by the original authors of the scale. A comparison of the models was obtained by looking at fit indices and factor loadings. Acceptable values for the model fit indices were: Comparative Fit Index (CFI) ≥ 0.95; Tucker–Lewis Index (TLI) ≥ 0.95; Root Mean Square Error of Approximation (RMSEA) ≤ 0.06; Standardized Root Mean Squared Error (SRMR) ≤ 0.08 [38]. Regarding missing data, no imputations were made, and analyses were conducted with the existing data. 

## 3. Results

### 3.1. Translation

An expert group (two geriatricians, an associate professor in geriatric nursing, and a cardiologist experienced in the translation of a frailty measurement) was created. A medical doctor and a journalist, both fluent in Swedish and English, translated the TFI into Swedish. The expert group discussed the translations. There was a pragmatic attitude in order to find the most reasonable wordings for older adults. Two back translations from Swedish were done, one to English by a geriatrician and one to Dutch by a senior researcher in medical sociology. After harmonization of the translations by the expert group, the first version of the TFI was tested on two geriatricians, three geriatric nurses, two healthy older persons, and five patients at a geriatric inpatient ward with cognitive debriefings. After the assessment, they were asked open questions regarding the design and their overall experience of the test. Concerns were raised regarding the three-answer alternatives for items 9–11, what distinguishes a “yes” from “sometimes”. The Swedish word for disease and disorder, “sjukdom”, is the same, and there was, therefore, some confusion among the patients when they were asked about diseases and conditions (part A, question 8). In the final version, the question is, therefore, only about diseases. Another question that was discussed at several steps of the process was how to describe an “important, intimate relationship” (part A, question 9). It was changed to “close” relationship (Swedish “nära”) since “intimate” could indicate a sexual relationship in Swedish. Several times during the translation process, there were suggestions about adding a third alternative for the question about sex. 

### 3.2. Descriptive Results

There were 315 participants included in the study. The mean age was 83.3 years, and 57.8% (*n* = 182) were women. The participants’ background characteristics and answers to part A of the TFI (determinants of frailty) are found in Table 1. The prevalence of frailty, defined as a total score of five or more, was 68% (*n* = 216). The mean TFI score was 6.5 (standard deviation (SD) 3.1); details of how the TFI questions were answered are found in Table 2. There was a high number of missing data for item 2, the question regarding weight loss (*n* = 39, 12.4%). Otherwise, the amount of missing data was rather low, ranging from 0.3% to 3.5% per item (Table 2). 

### 3.3. Reliability

#### 3.3.1. Inter-Item Correlations

The correlation matrix of the inter-item correlations for the 15 items of part B is presented in Table 3. For all three domains, there was a rather high proportion of weak correlations; the range was −0.04 to 0.55. Items one (overall health) and two (weight loss) were uncorrelated or had weak correlations with the majority of the other items of the physical domain. All items in the psychological domain were correlated, although item nine (memory problems) had rather weak correlations with the other items. In the social domain, item thirteen (living alone) did not correlate to item fourteen (missing having people around) and had a weak correlation to item fifteen (social support). 

#### 3.3.2. Internal Consistency

The reliability coefficients for the total scale were KR-20 0.69 and McDonald’s ω 0.72. For the subscales, the KR-20/McDonald’s ω was 0.61/0.63 for the physical domain, 0.65/0.68 for the psychological domain, and 0.43/0.45 for the social domain, respectively. According to the item–total correlation analysis, item 13 (if you live alone or not) could be removed along with an increase in the KR-20 reliability coefficient from 0.69 to 0.70 and the McDonald’s ω from 0.72 to 0.73 for the total scale.

#### 3.3.3. Test–Retest Reliability

Thirty-nine participants were tested by the same research nurse 14 (± 2) days after the first assessment. The average kappa value was 0.70, with some variation at the item level. Items 2, 5, 6, and 13 had kappa values over 0.8 (almost perfect, according to Landis and Koch), and items 12 and 14 had the lowest values, 0.46 and 0.43 (moderate) [29]. The percental agreement between the individual items ranged from 72.2% (item 14) to 100% (items 2 and 13).

### 3.4. Validity

#### 3.4.1. Concurrent Validity

The prevalence of frailty, measured by the Frailty Phenotype, was 51%. Of the 315 participants, only 252 (80%) were included in the analysis of the correlation between the TFI and the Frailty Phenotype. Due to the COVID-19 pandemic, some participants, *n* = 24 (7.6%), were only interviewed by telephone. The remaining participants, *n* = 39 (12.4%), were either habitually bedridden, in a wheelchair or. did not conclude the physical assessments due to different reasons; for example, some refused, and others were tired. The Pearson correlation coefficient (r) between the TFI and the Frailty Phenotype was 0.53 (*p* < 0.001). All but two participants (*n* = 313) had data on the Clinical Frailty Scale, and the prevalence of frailty was 55% with a correlation (r) of 0.46 (*p* < 0.001) with the TFI. 

#### 3.4.2. Construct Validity

The correlations between 14 of the 15 items of the TFI and their respective alternative measures are presented in Table 4. All items except two (items 2 and 5) correlated as expected and had significant correlations with their related measures. Ten items had moderately strong (0.30–0.60) correlations. 

#### 3.4.3. Structural Validity

In the one-factor model, the physical domain items 5 and 6, the psychological domain item 9, and the social domain item 13 presented factor loadings under 0.4 on the single factor. The fit indexes were: CFI 0.91, TLI 0.90, RMSEA 0.062 (90% CI 0.049–0. 076), and SRMR 0.12. The factor loadings for the three-factor model were similar to the loadings in the one-factor model, as presented in Figure 1. However, the three-factor model fitted the data better than the unidimensional structure. The fit indexes were: CFI 0.99, TLI 0.98, RMSEA 0.026 (90% CI 0.000–0.045), and SRMR 0.095. The physical and psychological domains were strongly correlated, while the social domain had weak correlations to the other two domains (Figure 1). 

## 4. Discussion

This study presents results from the translation and validation process of the Swedish version of the TFI in a population of community-dwelling older persons with multimorbidity and complex healthcare needs. The reliability of the Swedish TFI is evidenced by substantial test–retest stability and adequate internal consistency. The inter-item correlations were mostly classified as weak, which is acceptable but could be improved. However, lower inter-item correlations are expected for multidimensional constructs in comparison to narrower, unidimensional constructs [39,40]. Internal consistency results (KR-20 0.69 and McDonald’s ω 0.72) were close to findings in earlier studies (Dutch 0.73, Portuguese 0.78, German 0.67, Brazilian 0.78, and Polish 0.74) [9,41,42,43,44]. The low internal consistency of the psychological and social domains was expected and has, in earlier studies, been accepted as a reflection of the low number of items for these domains (4 and 3, respectively) [9,30]. We noticed that in our study, the internal consistency of the physical domain was lower than in previous studies; the reason for this remains unclear. Item 13, if one lives alone or not, proved to be problematic, and its removal increased the internal consistency of the whole scale. Living alone is considered a risk factor for frailty. However, living with someone does not automatically imply adequate social support since a partner could have conditions making them unable to provide support. Societal differences in welfare and access to economic support, together with cultural differences in formal and informal caregiving, could also affect living circumstances [45]. We suggest that item 13 (if one lives alone or not) could be removed from the TFI.

The test–retest reliability in this study was found to be substantial for most items and was done after 14 ± 2 days. Frailty is, however, an unstable condition that changes over time; most often, the condition deteriorates, but improvement also occurs with or without medical attention [46]. The consideration here is to choose a time period long enough for the participants to not remember what they answered in the first measurement (i.e., avoid the carry-over effects) but short enough for the level of the construct to remain relatively unchanged. The time period of 14 days is a commonly recommended time period for the test–retest analysis [47]. This rather short period diminishes the risk of instability in frailty between the measure points. 

The construct validity of the Swedish TFI was evidenced by significant, weak to moderate correlations of individual items with corresponding alternative measures. However, only five items met the predetermined value of ≥0.4, while five others were close to 0.4. BMI did not correlate with item 2, and since BMI does not represent a change in weight, it is probably unsuitable as an alternative measure, as recognized previously [43]. The correlations between the TFI and both the other frailty measures were not considered strong enough (>0.6) to provide evidence for concurrent validity. This was expected due to the differences in theoretical background. Similar results were reported in a sample (*n* = 856) of Spanish, community-dwelling older adults [15]. Another possible explanation for the low correlation with the Frailty Phenotype is that some participants were unable to perform the necessary physical measurements for the Frailty Phenotype. The correlations could, therefore, have been lowered due to a restricted sample. 

Regarding structural validity, there have been few studies of factor analysis of the TFI [15,16]. Lin et al. performed CFA when they studied the Taiwanese version of the TFI and found a good fit for a three-factor model [16]. Vrotsou et al. used both a three-factor model and a single-factor model when they studied the Spanish version of the TFI, where the first model showed acceptable model fit indices, suggesting a good representation of the studied data [15]. In the present study, the three-factor model showed better model fit than the one-factor model. However, several items with unacceptable loadings were observed, which Vroutso et al. also described [15]. However, if the items with low factor loadings were removed, the remaining items might be too few to measure frailty, and important information on possible opportunities for intervention might be lost. The strong correlation of the physical domain to the total score suggests that the answers to the physical frailty items were the most important for the total score in this study.

The mean age and prevalence of frailty were higher in our study compared to earlier studies of the TFI in community-dwelling populations [10]. This is probably a reflection of the inclusion criteria where individuals with multimorbidity (three or more diagnoses according to ICD-10) and high healthcare consumption (three or more visits to the ED in the past 18 months) were included, as described in the methods section. Only one study reports a higher prevalence of frailty in a population of older individuals living at a long-term care facility [48]. An interesting finding from this study is that 23% of the participants answered that they did not have two or more chronic conditions when, in fact, the inclusion criteria were three or more diagnoses in the ICD-10. This could perhaps partly be explained by cognitive deficits or how the older person defines or understand the term chronic condition.

The strength of the TFI is that it includes important aspects of frailty in only fifteen items. This pragmatic approach is highly relevant as it makes the assessment quick and does not fatigue the patients. Based on the findings in this study, the TFI can be recommended as a screening tool for frailty in older individuals with multimorbidity.

Concerns were raised regarding the three answer alternatives for items 9–11 and 14 (memory problems, depressive symptoms, anxiety, and social relations). Some older persons involved in the cognitive debriefings in the translation process found it hard to choose between the three alternatives. The expert group, however, did not want the Swedish version to be different from the original in this aspect. A dichotomization of the answer alternatives for items 9–11 and 14 could be considered in an eventual revision of the TFI. In the future, qualitative studies could further explore this subject.

Some limitations need to be addressed. Some missing data were expected due to the study population being old and at risk of cognitive decline and tiredness during the assessments. The amount of missing data on item 2 (weight loss), *n* = 39, was, however, unexpected. Additional analyses showed that the group with missing data had a significantly lower cognitive ability than those with data on weight loss. This could be interpreted as recollection bias. The TFI is characterized by a subjective measurement of frailty. This can be considered a limitation of the instrument; however, earlier studies have shown that self-reported information on frailty, both for the TFI and the frailty phenotype, has good predictive validity of health outcomes [49,50]. The predictive ability of the TFI for adverse outcomes (disability, lower quality of life, death) was, however, not explored for this study population, characterized by multimorbidity and complex needs of healthcare, and therefore requires further studies. Strengths include that data were collected by experienced research nurses with training in Good Clinical Practice, and face-to-face interviews ascertained a high response rate. Another strength was that the translation process followed robust guidelines and was documented in a structured process.

## 5. Conclusions

The Swedish version of the TFI is an adequately valid and reliable instrument for frailty screening in a population of older, community-dwelling people with multimorbidity. Results from this study suggest a potential for improvement, items 2 and 13 could be removed or rephrased, and a dichotomization of answer alternatives for items 9, 10, 11, and 14 could increase clarity and simplify the scale.

## Figures and Tables

**Figure 1 healthcare-11-02309-f001:**
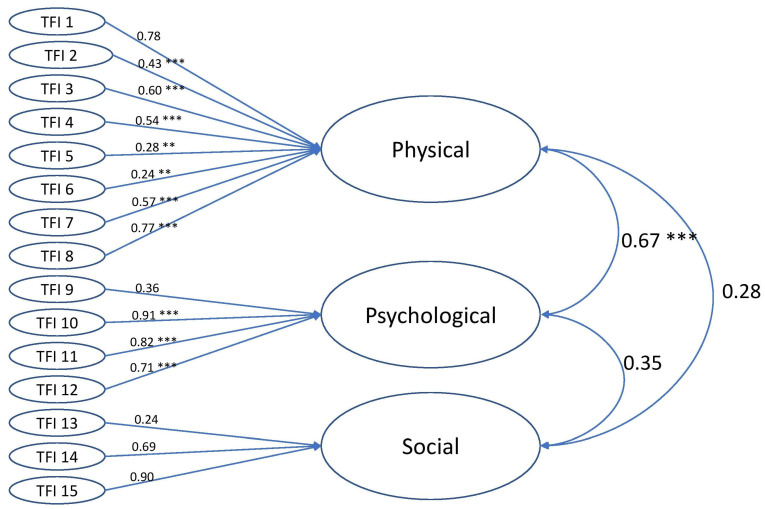
The three-factor CFA. Diagram over the three-factor model with loading factors and correlations between factors. Significance levels; ** *p* < 0.01, *** *p* < 0.001.

**Table 1 healthcare-11-02309-t001:** Participants’ characteristics and results of the Tilburg Frailty Indicator part A.

TFI Part A—Questions on Determinants of Frailty (Participants That Have Answered the Question)	*n* (%) Total = 315
Sex (women) (315)	182 (57.8)
Marital status (315)	
Married/Co-living	141 (44.8)
Widow-/er	121 (38.4)
Divorced	38 (12.1)
In a relationship but not living together	9 (2.9)
Unmarried	7 (2.2)
Born in Sweden (314)	284 (90.2)
Education (310)	
Primary school or less	188 (59.7)
High School	17 (5.5)
University, without degree	8 (2.5)
University with degree	59 (18.7)
Other	1 (0.3)
Income (307)	
<10,000 SEK *	21 (6.7)
10,000–15,000 SEK	112 (35.6)
15,001–20,000 SEK	57 (18.1)
20,001–30,000 SEK	53 (16.8)
30,001–45,000 SEK	25 (7.9)
≥45,001 SEK	7 (2.2)
Do not know	32 (10.2)
Lifestyle (304)	
Healthy	192 (61)
Not healthy/not unhealthy	79 (25.1)
Unhealthy	33 (10.5)
Multimorbidity (≥2 chronic diseases) (309)	
Yes	237 (75.2)
No	72 (22.9)
Life events during the past year	
Death of loved one (310)	97 (30.8)
Serious illness self (311)	102 (32.4)
Serious illness loved one (312)	90 (28.6)
Divorce (311)	5 (1.6)
Accident (312)	4 (1.3)
Crime (312)	25 (7.9)
Satisfaction of residence (Answer yes) (311)	273 (86.7)

* SEK = Swedish krona. SEK 100 is around EUR 9.5 (9.2–9.7 in 2022).

**Table 2 healthcare-11-02309-t002:** Results of the frailty assessment using the Tilburg Frailty Indicator part B.

Part B-The 15 Items of the Frailty Assessment (Participants That Have Answered the Question)	Answer *	*n* (% of Total 315) Mean (SD)
Physical frailty		
1 Physical healthy (309)	No	169 (53.7)
2 Weight loss (276)	Yes	22 (6.9)
3 Difficult in walking (309)	Yes	198 (62.9)
4 Difficult in balance (309)	Yes	188 (59.7)
5 Poor hearing (311)	Yes	152 (48.4)
6 Poor vision (310)	Yes	163 (51.7)
7 Weak hand strength	Yes	104 (33)
8 Physical tiredness (311)	Yes	195 (61.9)
Psychological frailty		
9 Problem with memory (310)		
	Yes	84 (26.7)
	Sometimes	126 (40)
	No	100 (31.7)
10 Feeling down (310)		
	Yes	90 (28.6)
	Sometimes	79 (25)
	No	141 (44.8)
11 Feeling nervous/anxious (311)		
	Yes	81 (25.7)
	Sometimes	53 (16.8)
	No	177 (56.1)
12 Cope with problems (301)	Yes	229 (72.7)
Social frailty		
13 Living alone (315)	Yes	169 (53.7)
14 Miss social relations (306)		
	Yes	83 (26.3)
	Sometimes	65 (20.6)
	No	158 (50.2)
15 Enough social support (308)	Yes	246 (78)
Frailty score		
Physical frailty (range 0–8) (309)	3.8 (1.9) mean (SD)	
Psychological frailty (range 0–4) (309)	1.5 (1.3) mean (SD)	
Social frailty (range 0–3) (308)	1.2 (1.0) mean (SD)	
Total frailty (range 0–15) (304)	6.5 (3.1) mean (SD)	
Total score ≥ 5 = FRAIL (298)	216 (68.6) *n* (%)	

* For items 1–8, 13, and 15 with two answer alternatives (Yes/No), only one alternative is presented. For items 9–11 and 14 with three answer alternatives, all three alternatives are presented.

**Table 3 healthcare-11-02309-t003:** Correlation matrix of the inter-item correlations of the TFI.

Items of TFI	1	2	3	4	5	6	7	8	9	10	11	12	13	14	15
Physical frailty															
1 Physical health	—														
2 Weight loss	0.105	—													
Difficulties due to:															
3 Walking	0.318 **	−0.008	—												
4 Balance	0.145 *	0.076	0.353 **	—											
5 Hearing	0.106	0.070	0.125 *	0.174 **	—										
6 Sight	0.084	0.026	0.122 *	0.190 **	0.150 **	—									
7 Hand strength	0.165 **	0.066	0.210 **	0.267 **	0.030	0.149 **	—								
8 Physical tiredness	0.428 **	0.138 *	0.339 **	0.254 **	0.195 **	0.093	0.194 **	—							
Psychological frailty															
9 Memory	0.061	0.059	0.073	0.142 *	0.186 **	0.022	0.017	0.081	—						
10 Depression	0.378 **	0.104	0.154 **	0.170 **	0.118 *	0.057	0.232 **	0.324 **	0.236 **	—					
11 Anxiety	0.296 **	0.116	0.079	0.148 **	0.007	0.066	0.292 **	0.295 **	0.157 **	0.549 **	—				
12 Coping	0.284 **	0.142 *	0.123 *	0.166 **	0.091	0.057	0.163 **	0.213 **	0.255 **	0.322 **	0.372 **	—			
Social frailty															
13 Living alone	0.100	0.008	0.045	0.050	0.070	0.119 *	0.075	0.041	−0.041	0.037	0.028	−0.004	—		
14 Missing people	0.171 **	0.169 **	0.101	0.063	0.084	0.047	0.065	0.062	0.049	0.167 **	0.127 *	0.121 *	0.095	—	
15 Support	0.178 **	0.090	0.016	0.102	0.032	0.057	0.087	0.089	0.083	0.194 **	0.171 **	0.103	0.152 **	0.380 **	—

* *p* < 0.05, ** *p* < 0.01.

**Table 4 healthcare-11-02309-t004:** Correlation between items and their alternative measure.

TFI Item	Alternative Measure	*n*	r	*p*
1. Do you feel physically healthy?	International Physical Activity Questionnaire (IPAQ) [28] *	302	−0.177	0.002
2. Have you lost a lot of weight recently without wishingto do so?	BMI	282	−0.051	0.419
Do you experience problems in your daily life due to:				
3. difficulty in walking?	Walking Speed (Short Physical Performance Battery, SPPB) [29]	251	−0.375	<0.001
4. difficulty maintaining your balance?	Balance Test (Short Physical Performance Battery, SPPB) [29]	268	−0.374	<0.001
5. poor hearing?	Question on hearing-Can you make telephone calls: Yes/Yes with aid/No	301	−0.001	0.981
6. poor vision?	Question on sight-Can you read the words in the questionnaire? Yes, sharp/Yes, most/Yes, some/Only little/No	285	−0.207	<0.001
7. lack of strength in your hands?	Grip strength measured with a dynamometer	285	−0.548	<0.001
8. physical tiredness?	Question on exhaustion from the Center for Epidemiological Studies Depression Scale (CES-DS) [22]	305	0.444	<0.001
9. Do you have problems with your memory?	Montreal Cognitive Assessment (MoCA) [33]	276	−0.319	<0.001
10. Have you felt down during the last month?	Hospital Anxiety and Depression Scale-Depression subscale (HADS-D) [32]	302	0.399	<0.001
11. Have you felt nervous or anxious during the last month?	Hospital Anxiety and Depression Scale-Anxiety subscale (HADS-A) [32]	295	0.466	<0.001
12. Are you able to cope with problems well?	Pearlin Mastery Scale (PMS) [34]	265	−0.441	<0.001
13. Do you live alone?	Not included in the analysis.	X		
14. Do you sometimes miss having people around you?	Question on loneliness - Does it happen that you are troubled by loneliness? Almost always/Often/Sometimes/Almost never [35]	306	−0.419	<0.001
15. Do you receive enough support from other people?	Oslo 3-item social support scale (OSSS) [36]	272	−0.344	<0.001

* Numbers in brackets are references.

## Data Availability

The dataset used during the current study is available from the corresponding author upon reasonable request and insofar as it is in accordance with Swedish law. The data are not publicly available due to their sensitive nature.

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
