# Peer review of "Translation and Validation of the Swedish Version of the Tilburg Frailty Indicator"

_healthcare, 2023, doi:10.3390/healthcare11162309_

Round 1
Reviewer 1 Report
It's a very well written manuscript with appropriate aim, sound methods and conclusions. The findings add to the exisitng body of knowledge in the field and allow improvement of the original Tilburg Frailty Indicator (TFI).
My only suggestion would be expand abbreviations in the text of the aim of the study and though I may seem old-fashioned to limit the use of abbreviations throughout the text.
Wishing all the best to the author team!
Reviewer 2 Report
The paper describes the validation of the TFI in a sample of Swedish individuals. Methods and statistics were described adequately. The power of the study is sufficient. Results are convincing and limitations are discussed well.
Reviewer 3 Report
The work is very interesting and the authors have written in detail all the methodological choices and the results obtained. The translation was reportedly accurate and correctly placed in the Swedish context. However, the research design has some weaknesses:
- An element of subjectivity has been introduced in identifying the methods of measuring the elementary dimensions; this may have had a significant influence on the final result.
- The research was conducted on a sample collected with a very restrictive inclusion criterion which led to a deterioration of the information collected, even if in reality this measuring device will be applied precisely to this type of person in conditions of fragility.
For these reasons, the result on the evaluation of the device is not entirely satisfactory.
Alternatively (but it is clear that this would involve a lot of work because all the analyzes would have to be repeated) the authors could enlarge the sample and also include other subjects in less frail conditions to fill some gaps in the results. In fact, the device for measuring fragility should be universal and provide an assessment even in cases of absence of fragility and for this reason it can also be tested on this type of patient. However, if we admit that the performance is partly due to these choices, we can rethink the result in more positive terms and this should be included in the conclusions which are really too brief at the moment.
I don't want to say anything about english, just re-read to avoid oversights like god instead of good a line 314
Reviewer 4 Report
line 86 change instrument --> instruments
line 87, add "and its underlying factors."
line 119, "determinants" of what ?
Change FP to Fried Frailty Index and indicate that a modified FFI was used (i.e, IPAQ-SF replaced the Fried question on physical activity)
Inclusion of wheelchair bound older adults as frail is unacceptable because not all wheelchair bound people are frail and in fact may be very robust. Need to revise data with the exclusion of wheelchair bound subjects.
Need minor grammatical revisions
Reviewer 5 Report
HEALTHCARE - 2075775
Brief Summary:
This was a methodologic study examining the systematic translation of a frailty scale (Tilburg Frailty Indicator, TFI) from Dutch to Swedish, and then performing tests of reliability and validity for a sample of community-dwelling older people.
This is generally an extremely well thought out and above average manuscript.
General comments to be addressed
Line 86: the first word of this line should be plural "instruments". Also in that same line "factor analysis of an existing instrument (add the word "an")
Tables: general comments: you can make these much easier to read and use less space if you would left align the options under items 9-15. Also, readers look to the column heading to understand the numbers listed and when you mix in mean and standard deviations you should include that in the column heading not just the line for the item.
Table 1: The column is labeled n and % but the numbers in the column do not all seem to be this. For example, age is stated to be mean and s.d. in the first line so this does not match what is in the column. Also, it these numbers are correct how do you account for those under age 75 (your selection criterion) if some are obviously two sd below the mean? Also, the multimorbidy is listed as 2 or more chronic diseases, but the selection criterion was 3 or more. Please correct and make your table match your selection criteria or explain in footnotes for the table why these are not the same.
Table 2: Item 15 is listed as only yes but not sometimes or no like all the other items in this category. Please explain with a foot note or correct.
Table 3: add to the title: “for the TFI”. This table could be consolidated with a more typical presentation such as may be found at this link https://apastyle.apa.org/style-grammar-guidelines/tables-figures/sample-tables Please note that some kind of item description other than just Item 1, would be very helpful to the reader.
Line 236-234: If the scale measures three different concepts, then one would expect inter-item correlations to be low except within those three concepts. If all items had significant correlation with every other item, then scale has a lot of redundancy and could possibly be reduced in number. However, frailty is NOT a single concept as your research clearly demonstrates. Also remember that test-retest reliability is assuming a stable trait(s) which again, frailty is not stable, it continues to change. So be careful in how you interpret your results on this in the discussion.
Line 250-261: It is curious to me that in a sample with suspected frailty only 51% were found by the TFI to be frail. In your future studies seriously consider studying those who did NOT test as frail. What about them is different.
Table 4: in column 2 you have a number in brackets, please note what this is or omit. I think it may be the item number of the scale but you do not say this anywhere. You list BMI but do not mention this anywhwere else except in line 135 as an adjuster for the weakness hand grip. It was not used in your analysis so I do not understand why this is in the table. It does not correlate with the TFI item of weight loss, only weight does. Please explain and correct.
Line 309-312: If the some were unable to perform the FP measurements, then are those measurements valid to detect frailty? Maybe different measures are needed such as more practical items for your tool (can you open a new jar, do you have difficulty opening car doors or house doors, etc.).
Line 320-21 does not make sense as written “The propose questions about the construction of the instrument.” Omit? Please fix.
See above.
Round 2
Reviewer 5 Report
All prior concerns have been addressed.